# High Incidence of SARS-CoV-2 Variant of Concern Breakthrough Infections Despite Residual Humoral and Cellular Immunity Induced by BNT162b2 Vaccination in Healthcare Workers: A Long-Term Follow-Up Study in Belgium

**DOI:** 10.3390/v14061257

**Published:** 2022-06-09

**Authors:** Bas Calcoen, Nico Callewaert, Aline Vandenbulcke, Winnie Kerstens, Maya Imbrechts, Thomas Vercruysse, Kai Dallmeier, Johan Van Weyenbergh, Piet Maes, Xavier Bossuyt, Dorinja Zapf, Kersten Dieckmann, Kim Callebaut, Hendrik Jan Thibaut, Karen Vanhoorelbeke, Simon F. De Meyer, Wim Maes, Nick Geukens

**Affiliations:** 1Laboratory for Thrombosis Research, KU Leuven Campus Kulak Kortrijk, 8500 Kortrijk, Belgium; bas.calcoen@kuleuven.be (B.C.); aline.vandenbulcke@kuleuven.be (A.V.); karen.vanhoorelbeke@kuleuven.be (K.V.); simon.demeyer@kuleuven.be (S.F.D.M.); 2AZ Groeninge Hospital, Department of Laboratory Medicine, 8500 Kortrijk, Belgium; kimcallebaut@hotmail.com (K.C.); nico.callewaert@azgroeninge.be (N.C.); 3Laboratory of Virology and Chemotherapy, Translational Platform Virology and Chemotherapy, Department of Microbiology, Immunology and Transplantation, KU Leuven Rega Institute, 3000 Leuven, Belgium; winnie.kerstens@kuleuven.be (W.K.); thomas.vercruysse@kuleuven.be (T.V.); hendrikjan.thibaut@kuleuven.be (H.J.T.); 4PharmAbs, the KU Leuven Antibody Center, KU Leuven, 3000 Leuven, Belgium; maya.imbrechts@kuleuven.be (M.I.); nick.geukens@kuleuven.be (N.G.); 5Laboratory of Virology, Molecular Vaccinology and Vaccine Discovery, Department of Microbiology, Immunology and Transplantation, KU Leuven Rega Institute, 3000 Leuven, Belgium; kai.dallmeier@kuleuven.be; 6Laboratory for Clinical and Epidemiological Virology, KU Leuven Rega Institute, 3000 Leuven, Belgium; johan.vanweyenbergh@kuleuven.be (J.V.W.); piet.maes@kuleuven.be (P.M.); 7Department of Microbiology, Immunology and Transplantation, KU Leuven, 3000 Leuven, Belgium; xavier.bossuyt@uzleuven.be; 8Department of Laboratory Medicine, University Hospitals Leuven, 3000 Leuven, Belgium; 9Institut für Experimentelle Immunologie, EUROIMMUN Medizinische Labordiagnostika AG, 23552 Lübeck, Germany; d.zapf@euroimmun.de (D.Z.); k.dieckmann@euroimmun.de (K.D.)

**Keywords:** SARS-CoV-2, BNT162b2 vaccine, long-term monitoring, healthcare workers, breakthrough infection, variants of concern

## Abstract

To mitigate the massive COVID-19 burden caused by severe acute respiratory syndrome coronavirus 2 (SARS-CoV-2), several vaccination campaigns were initiated. We performed a single-center observational trial to monitor the mid- (3 months) and long-term (10 months) adaptive immune response and to document breakthrough infections (BTI) in healthcare workers (*n* = 84) upon BNT162b2 vaccination in a real-world setting. Firstly, serology was determined through immunoassays. Secondly, antibody functionality was analyzed via in vitro binding inhibition and pseudovirus neutralization and circulating receptor-binding domain (RBD)-specific B cells were assessed. Moreover, the induction of SARS-CoV-2-specific T cells was investigated by an interferon-γ release assay combined with flowcytometric profiling of activated CD4^+^ and CD8^+^ T cells. Within individuals that did not experience BTI (*n* = 62), vaccine-induced humoral and cellular immune responses were not correlated. Interestingly, waning over time was more pronounced within humoral compared to cellular immunity. In particular, 45 of these 62 subjects no longer displayed functional neutralization against the delta variant of concern (VoC) at long-term follow-up. Noteworthily, we reported a high incidence of symptomatic BTI cases (17.11%) caused by alpha and delta VoCs, although vaccine-induced immunity was only slightly reduced compared to subjects without BTI at mid-term follow-up.

## 1. Introduction

Back in December 2019 in Wuhan, China, multiple severe lower respiratory tract infections (later renamed as coronavirus disease 19; COVID-19) were reported. The causative micro-organism was identified as a novel β-coronavirus [1]. This pathogen was named “severe acute respiratory syndrome coronavirus 2” (SARS-CoV-2) [2] since it enters the human cell via a similar mechanism as observed for the SARS-CoV virus, the pathogen responsible for the earlier SARS outbreak in China [3]. In short, the viral receptor-binding domain (RBD) of the spike (S) protein [4] interacts with the human angiotensin receptor enzyme 2 (ACE2) [5] to enter epithelial cells followed by intra-cellular replication. Eventually, SARS-CoV-2 rapidly spread across the globe and led to the largest pandemic of the digital age [6,7].

Initially, never-before-seen large-scale socio-economic measures were taken by governments to mitigate further spreading of SARS-CoV-2, including a hard lockdown of societies and the obligation to wear face masks. Additionally, both curative and prophylactic strategies were urgently designed to prevent collapse of healthcare systems due to the massive SARS-CoV-2-related impact on human health [8]. In this context, different SARS-CoV-2 vaccines were developed and approved for use in the general adult population at an unprecedented speed. These vaccines included either nucleic acid-based (e.g., BNT162b2 or Comirnaty^®^ [9,10], mRNA-1273 or Spikevax^®^ [11,12]) or adenovector-based (e.g., Vaxzevria^®^ [13], COVID19 vaccine Janssen^®^ [14,15]) vaccines.

Prior to the COVID-19 pandemic, no detailed information was available concerning nucleic acid-based vaccine-induced immunity and response sustainability compared to the more established vector- and protein-based vaccines. Hence, the large (inter-)national vaccination campaigns presented a unique opportunity to study the nucleic acid-based vaccine-induced immune reaction and longevity. Indeed, a myriad of reports were published describing both SARS-CoV-2-specific B cell or T cell reactions after partial (i.e., single dose) or full vaccination (i.e., two doses) with either the BNT162b2 or mRNA-1273 vaccine within several study groups (e.g., healthcare personnel and immunocompromised patients) up to 6 months after vaccination [16,17,18,19,20,21,22]. However, long-term documentation of integrated humoral and cellular immunity is scarce, in particular in healthy SARS-CoV-2-naive subjects. If present, most of these reports primarily focused on monitoring serological responses only [16,17,18,19]. Other reports that did perform long-term follow-up monitoring mostly focused on a specific patient group, such as SARS-CoV-2 convalescent [20] or immunocompromised patients [21]. The interim analysis of this study is available as a preprint on the medrxiv server [22].

Unfortunately, several SARS-CoV-2 variants of concern (VoCs) have emerged over time [23] and caused significant new infection waves [24,25,26] even in countries with a very high vaccination rate. This questions whether the vaccine-induced activity response is maintained against these new VoCs. Indeed, these VoCs acquired one or multiple point mutations in the S protein and could therefore escape the vaccine-induced immunity raised against ancestral Wuhan-type SARS-CoV-2. Accordingly, the BNT162b2 vaccine phase II/III trials, executed before the VoCs emergence, reported a much lower incidence of breakthrough infections (BTI) compared to several real-world reports [27,28]. It is noteworthy that several case reports have published conflicting data on disease severity in people that experienced a BTI after vaccination with BNT162b2 [29,30]. Finally, it has not yet been clearly investigated whether the vaccine-induced immune reaction and sustainability in subjects experiencing symptomatic BTI (i.e., BTI group) is different compared to individuals that did not develop a BTI (i.e., non-BTI group).

To address above-mentioned issues, we prospectively assessed the BNT162b2 vaccine-elicited immune responses within healthy SARS-CoV-2-naive healthcare workers in Belgium for up to 10 months in a real-world setting with multiple VoCs emerging.

## 2. Materials and Methods

### 2.1. Study Design

#### 2.1.1. Recruitment

In January 2021, SARS-CoV-2-naive healthcare workers from the supraregional AZ Groeninge hospital (Kortrijk, Belgium) able to receive the BNT162b2 vaccine were contacted via the hospital’s newsletter for voluntary enrollment in this trial. Healthcare workers were randomly included (probability sampling) after signing an informed consent form. Participants were excluded if they met at least one of the following criteria: ongoing severe acute infection of any kind, pregnancy, history of a laboratory-proven immunodeficiency (e.g., primary immunodeficiency), chronic treatment with immunomodulatory agents (e.g., anti-TNF-α agents, corticosteroids), documented earlier natural infection with SARS-CoV-2 or positive serology (including either anti-S IgG, anti-S IgA or anti-nucleocapsid (N) IgG) found at baseline (i.e., prior to vaccination) sampling.

#### 2.1.2. Sampling

A schematic overview of the sampling procedure is shown in Figure 1a. No longer than 24 h before prime vaccination, a sampling moment was planned to establish a baseline immune profile (t_pre_). BNT162b2 administration was performed according to the recommendations of both the manufacturer and the Belgian Superior Health Council. A second sampling was performed three months (+/− 9 days) post prime vaccination (i.e., mid-term follow-up; t_3m_). For each participant, the last sampling was taken at 10 months (+/− 30 days) post prime vaccination (i.e., long-term follow-up test; t_10m_). In case a fully vaccinated participant (i.e., participant that received two doses of the BNT162b2 vaccine) presented with suggestive symptoms for SARS-CoV-2 infection (e.g., dry cough, fever and/or headache) or had a prolonged high-risk contact and also tested positive for SARS-CoV-2 via reverse transcriptase–polymerase chain reaction (RT-PCR), this participant was defined to have a BTI. In this setting, that specific participant was recalled for an additional sampling as soon as possible (t_BTI_). At t_BTI_, apart from venous blood (Figure 1a), a nasopharyngeal swab was also taken. Additionally, the participant was asked to fill in an in-house developed disease severity questionnaire (Appendix A). Blinding was guaranteed by pseudonymization of all samples at any time.

#### 2.1.3. Peripheral Blood Mononuclear Cells (PBMC) Isolation and Serum Collection

Blood from three K_2_-EDTA tubes (BD Vacutainer^®^, BD, Mississauga, ON, Canada) was used to isolate PBMCs via Ficoll-mediated (lymphoprep^TM^, STEMCELL technologies, Grenoble, France) density gradient isolation with SepMate tubes (STEMCELL technologies, Grenoble, France). PBMCs were diluted in a freezing medium (fetal calf serum with 10% DMSO), divided into aliquots and stored in liquid nitrogen after a controlled cooling procedure. When needed, PBMC aliquots were thawed at 37 °C for 2 min and washed two times in dPBS (Gibco^TM^ Life Sciences, Thermo Fisher Scientific, Waltham, MA, USA) with a centrifugation step at 300 g for 7 min at 4 °C after each washing. Serum was collected from the SST^TM^ II Advance tubes (BD Vacutainer^®^, BD, Mississauga, ON, Canada) and stored in aliquots at −20 °C.

### 2.2. Serological Parameters

#### 2.2.1. Anti-S IgA and IgG Assay

Serum anti-S IgA antibodies were measured with the Anti-SARS-CoV-2 IgA enzyme immunoassay from EUROIMMUN (Lübeck, Germany) on an ETI-Max 3000 instrument from DiaSorin (Saluggia, Italy). Following the instructions from the manufacturer, samples with a cut-off index greater or equal to 1.1 were labeled as positive. Serum anti-S IgG titers were measured with the VIDAS SARS-CoV-2 IgG (9COG) enzyme immunoassay from Biomérieux (Marcy-l’Etoile, France) on a VIDAS 3 instrument from the same manufacturer. For these assays, recombinant S1 domain of the spike protein of the Wuhan-Hu-1 SARS-CoV-2 isolate was used. According to the manufacturer’s instructions, samples with a cut-off index greater or equal to 1.0 were considered positive.

#### 2.2.2. Anti-RBD IgG Assay

Enzyme-linked immunosorbent assay (ELISA) plates (Corning Costar; cat. Nr. 3590) were coated overnight with ancestral Wuhan His_6_-tagged RBD (Arg_319_-Phe_541_, Sino Biologicals, cat. Nr. 40592-V08H, Houston, TX, USA). Plates were blocked for 2 h at room temperature (RT) using a blocking buffer (PBS + 1% BSA) and washed with a wash buffer (PBS + 0.002% Tween 80) six times. Serum samples (diluted: minimum 500-fold) were incubated for 2 h at RT (buffer PBS + 0.1% BSA + 0.002% Tween 80). After an additional washing step, goat antihuman (GAH) IgG conjugated with horseradish peroxidase (HRP) was added (1:5000 dilution) and incubated for 1 h at RT. Subsequently, washing was performed, and the plate was developed using o-phenylenediamine dihydrochloride (OPD; 0.4 g/L) and H_2_O_2_ (0.003%) in citrate buffer. After 30 min, the reaction was stopped with H_2_SO_4_ (4 M). The absorbance was measured at 492 nm, and the dose-response curve was analyzed by non-linear regression using GraphPad Prism 9.0.0 (Graph Software, San Diego, CA, USA). The assay was validated by measuring assay cut-off values for detection and quantification, accuracy, imprecision and dilutional linearity [32]. The limit of detection (LOD) for this assay was 160 BAU/mL. Sample concentrations were determined using two calibrators: (i) the WHO International Standard Serum (NIBSC refs 20–136) [33] or (ii) reference serum (kindly provided by Red Cross Flanders) that was bridged to the WHO standard.

#### 2.2.3. Anti-N IgG Assay

The presence of serum anti-N IgG antibodies was determined via the Anti-SARS-CoV-2-NCP (IgG) enzyme immunoassay from EUROIMMUN on an ETI-Max 3000 instrument from DiaSorin. Samples that had a cut-off index greater or equal to 1.1 were considered positive as recommended by the instructions from the manufacturer.

### 2.3. Functional Assessment of Vaccine-Induced Antibodies

#### 2.3.1. In Vitro RBD-ACE2 Binding Inhibition Assay

Determination of the in vitro capacity of the vaccine-elicited antibodies to inhibit the interaction between viral RBD and human angiotensin converting enzyme 2 (ACE2) receptor was performed using the EUROIMMUN SARS-CoV-2 NeutraLISA assay (cat Nr. EI 2606-9601-4) following the manufacturer’s instructions. For this assay, the RBD domain of the S protein of the Wuhan-Hu-1 SARS-CoV-2 isolate was used. Serum samples were diluted 1:5 in the sample buffer. A photometric measurement was made on a wavelength of 450 nm together with a reference wavelength of 620 nm. Semiquantitative results were generated by calculating a ratio of the extinction values of the sample over the mean extinction value of the blank (measured in duplicate) and were presented as percentage inhibition (% IH). As stated by the manufacturer, percentage inhibition values lower than 20 were defined as negative, values between 20 and 35 as borderline and values higher or equal to 35 as positive. Lot-specific control concentrates (positive and negative) included in the assay kit were used as assay references.

#### 2.3.2. Vesicular Stomatitis Virus (VSV) Pseudovirus Neutralization Assay

VSV S-pseudotypes were generated as described previously [34]. The different pseudotypes were generated using S expression plasmids of prototype B.1/D614G as before [34] or sourced from Invivogen for VoC Delta (Cat. No. plv-spike-v8). Briefly, HEK-293T and BHK-21J cells were transfected with prototype B.1/D614G, from here on referred to as D614G, and the VoC Delta S protein expression plasmids, respectively, and one day later infected with green fluorescent protein (GFP)-encoding VSVΔG backbone virus [35]. Two hours later, the medium was replaced by a medium containing anti-VSV-G antibody (I1-hybridoma, ATCC CRL-2700) to neutralize residual VSV-G input. After 26 h of incubation at 32 °C, the supernatants were harvested. To quantify neutralizing antibodies (nAbs), serial dilutions of serum samples were incubated for 1 h at 37 °C with an equal volume of S pseudotype VSV particles and inoculated on Vero E6 cells (SARS-CoV-2) for 19 h.

The percentage of GFP-expressing cells was quantified on a Cell Insight CX5/7 High Content Screening platform (Thermo Fischer Scientific) with Thermo Fisher Scientific HCS Studio (v.6.6.0) software. Neutralization IC50 values were determined by normalizing the serum neutralization dilution curve to a virus (100%) and cell control (0%) and fitting in GraphPad Prism (inhibitor vs. response, variable slope and four parameters model with top and bottom constraints of 100% and 0%, respectively).

### 2.4. Analysis of SARS-CoV-2-Specific T Cells

SARS-CoV-2-specific T cell activity was assessed via stimulation of the T cells using the EUROIMMUN Quan-T-Cell SARS-CoV-2 assay (cat Nr. ET 2606-3003) that was performed according to the manufacturer’s instructions adapted as described earlier [22]. First, 500 µL of heparinized whole blood was transferred into three different tubes (BLANK, COV2 and STIM) followed by a 20–24 h incubation step at 37 °C after inverting these tubes 6 times. The COV2 tube was coated with S1 domain antigens of SARS-CoV-2. Following incubation, all tubes were centrifuged at RT for 10 min at 700 g. Approximately 200 µL of heparinized plasma from each tube was pipetted into Eppendorf tubes and centrifuged again at RT for 10 min at 12,000 g. Finally, the supernatants were pipetted into cryovials and stored at −20 °C until measurement via the EUROIMMUN Quant-T-Cell ELISA (see Section 2.4.1). The remaining pellet containing the stimulated cellular fractions within the tubes was used for additional flowcytometry (see Section 2.4.2).

#### 2.4.1. Interferon γ (IFN-γ) ELISA

Specific SARS-CoV-2-induced IFN-γ release was determined via the EUROIMMUN Quant-T-Cell ELISA (cat Nr. EQ 6841-9601) according to the manufacturer’s instructions. A photometric measurement was performed at a wavelength of 450 nm with a reference measurement at 620 nm. For each tube, IFN-γ concentrations were determined using a standard curve that was fitted via GraphPad Prism (four parameters model without restrictions). Then, for each subject, the determined IFN-γ concentration from the unstimulated control (BLANK) was subtracted from the determined IFN-γ concentrations of both the stimulation control (STIM) and the SARS-CoV-2 stimulated condition (COV2). Lot-specific lyophilized calibrators and controls included in the assay kit were used as a standard.

#### 2.4.2. T Cell Phenotyping

The remaining stimulated cell pellets from the IFN-γ release assay tubes (see Section 2.4) were immediately resuspended in dPBS in a total volume of 500 µL. Next, a whole blood staining was performed on the reconstituted samples. In summary, T cell staining was performed on 150 µL of the reconstituted fractions [22]. Following incubation for 30 min at 4 °C, 3 mL of red blood cell (RBC) lysis buffer (BD FACS^TM^ lysing solution, BD Biosciences, Franklin Lakes, NJ, USA) was added for 5 min at RT to allow lysis of the RBC fraction. After extensive washing, the pellets were resuspended in 350 µL dPBS and immediately acquired on a flowcytometer (FACSVerse device, BD Biosciences, Franklin Lakes, NJ, USA).

T cells were selected via gating (Appendix A) on the CD3^+^ population and further divided into helper T cells (T_H_; CD4^+^/CD8^−^), circulating follicular helper T cells (T_CFH_; CD4^+^/CD8^−^/CXCR5^hi^) and cytotoxic T cells (T_C_; CD4^−^/CD8^+^). Membrane markers used to assess T cell activation were CD40L and CD69. Gates based on the fluorescence minus one (FMO) signal retrieved for each individual fluorochrome were added to define marker positivity. For each subject, besides the condition with SARS-CoV-2-specific antigens (Appendix A), an unstimulated negative (Appendix A) and a positive control condition (Appendix A) were available to respectively correct for background and to assess intrinsic cell functionality.

### 2.5. Quantification of Circulating RBD-Specific B Cells

Thawed PBMCs were stained with a selective B cell staining panel that can be retrieved in the interim analysis of this study [22]. After a final washing step, the pellets were resuspended in 300 µL dPBS and immediately acquired on a flowcytometer (FACSVerse device, BD Biosciences, Franklin Lakes, NJ, USA). Living B cells were selected from the PBMC pool via a CD3^−^/CD19^+^/Zombie^−^ gating strategy as described earlier [22].

Specific B cell reactivity against SARS-CoV-2 was assessed using ancestral Wuhan-type RBD-biotin (Arg_319_-Phe_541_, Sino Biologicals, cat. Nr. 40952-V27H-B) and PE-streptavidin as described by Imbrechts et al. [36]. For each sample, a negative control tube (without RBD-biotin) was included to correct for sample-specific background. The biological relevant cut-off of this assay was set at 0.01% of living B cells. Additionally, for each staining experiment, a sample with documented RBD-specific B cells was used as a positive control for quality assessment.

### 2.6. Viral Whole Genome Sequencing

RNA extraction was performed on nasopharyngeal swabs taken at t_BTI_, using the DEXR-15-LM96 kit for automated extraction (Diagenode, Seraing, Belgium) with 350 µL sample input. Extracted RNA was eluted from magnetic beads in 50 μL of UltraPure DNase/RNase-free distilled water. Following RNA extraction, cDNA was synthesized followed by multiplex PCR amplification using a modified version of the ARTIC V3 LoCost protocol with the Midnight primer set (1200 bp amplicons). The libraries were sequenced on a MinION using R9.4.1 flow cells (Oxford Nanopore Technologies, Oxford, UK) and MinKnow software (v21.02.1 for Windows, Oxford Nanopore technologies, Oxford, UK) The resulting fast5 reads were base called and demultiplexed using Guppy (v5.0.16 for Windows, Oxford Nanopore technologies, Oxford, UK) in super accuracy mode. Genome assembly was performed using the ARTIC bioinformatics pipeline v1.1.3, which entails adapter trimming and mapping to the reference strain Wuhan-Hu-1 (MN908947), as previously described [37].

### 2.7. Statistical Analyses

Statistical analysis was performed using Microsoft Excel (v365 for Windows, Microsoft Corporation, Birmingham, Alabama, USA) or GraphPad Prism (v9.0.0 for Windows, GraphPad Software, San Diego, CA, USA). Flowcytometric data were processed using FCS Express (v7.10.0007 research edition for Windows, De Novo Software, Los Angeles, CA, USA). Continuous variables were presented as mean ± standard deviation if the Shapiro–Wilk normality test was successful or as median ± interquartile range (IQR) if not. When median ± IQR yielded an interval with no relevant biological meaning, continuous variables were presented as the interval between quartile 1 and 3 (Q1–Q3). Confidence intervals (CI) of medians were calculated via the standard method described by Zar JH [38]. Discrete variables were shown as numbers with percentages between brackets. Raw data were screened for outlier values via the ROUT method with a Q-value of 1%. If there were outlier values present, these were excluded from subsequent analyses. Comparison between parameters was done using a Student’s t-test or one-way ANOVA with a post hoc correction for multiple comparisons via Tukey’s test and after an assessment of constant variance using Levene’s test if normality was met or with the respective non-parametric alternatives when there was no normality. Paired analyses were done if appropriate. Correlation between parameters was assessed via bivariate analysis and expressed via a Pearson determination coefficient (R^2^), a Pearson r or via a Spearman r if there was no normality, as well as via a multivariate principal component analysis (PCA). Sample size determination was performed by power analysis for all tests with significance level (alpha) set at 0.05 and power (1-beta) set at 0.80.

## 3. Results

### 3.1. Trial Characteristics and Exclusions

Eighty-four Caucasian SARS-CoV-2-naive healthcare workers were enrolled for this study. Their age ranged between 24 and 75 years with a median age of 41 ± 20 (95% CI: 36-48) and approximately half of the participants (55%) were women.

During the study, a total of nine participants dropped out between t_pre_ and t_10m_ and were excluded for all subsequent analyses. Eight of these participants dropped out between t_pre_ and t_3m_, while only one drop-out case was noted between t_3m_ and t_10m_. Furthermore, 62 of the remaining 75 participants (83%) did not report suggestive symptoms and had anti-N IgG titers below the cut-off index value on all sampling moments, confirming absence of SARS-CoV-2 infection (i.e., non-BTI group). The remaining 13 participants (17%) developed a (sub)clinical BTI within the period between receiving the second vaccine and the t_10m_ sampling timepoint (i.e., BTI group; Figure 1b).

### 3.2. Preservation of Functional In Vitro Neutralization over Time despite Waning Antibody Levels

To start, an extensive humoral vaccine response monitoring was performed and included (i) SARS-CoV-2-specific serology, (ii) a functional antibody assessment using both in vitro inhibition of the binding between viral RBD and human ACE2 and (iii) pseudovirus neutralization assays. A subset of these data taken at t_3m_ has already been published [22]. To reliably describe the evolution of the vaccine-induced humoral immunity over time, all data from participants with reported BTI were excluded and analyses shown in this paragraph are data of the non-BTI group (*n* = 62).

Firstly, for all 62 subjects, baseline titers for both anti-S IgG and anti-S IgA were below cut-off index values. Likewise, pre-vaccination anti-RBD IgG titers were undetectable (i.e., below assay-specific LOD). Interestingly, anti-S IgG, anti-S IgA and anti-RBD IgG titers all waned significantly (median t_10m_/t_3m_ ratio: 0.21 ± 0.14 with *p* < 0.001; median t_10m_/t_3m_ ratio: 0.60 ± 0.23 with *p* < 0.001; and median t_10m_/t_3m_ ratio: 0.08 ± 0.07 with *p* < 0.0001, respectively) between t_3m_ and t_10m_ (Figure 2). Furthermore, 3, 15 and 30 participants had no quantifiable anti-S IgG, anti-S IgA and anti-RBD IgG titers, respectively, at t_10m_ (data not shown). For both mid- and long-term follow-up, anti-S IgG, anti-S IgA and anti-RBD IgG antibody titers displayed significant correlation (Table 1—row 1–3 and Appendix A).

Next, functionality of the vaccine-induced antibodies was assessed via two methods. Firstly, the capacity of the antibodies to inhibit the interaction between ancestral Wuhan RBD and human ACE2 was assessed. Inhibition of the RBD–ACE2 interaction was below the assay-specific LOD at baseline but was clearly detectable in all but one subject at t_3m_ with a median of 93 ± 11% IH. Binding inhibition attenuated significantly (*p* < 0.001) at t_10m_ with a median of 44% IH and a Q1–Q3 interval of 20–68% IH. Moreover, 10 of the 62 participants showed borderline residual functionality and 14 subjects displayed no inhibition capacity anymore at t_10m_ (Figure 3a). Secondly, neutralization activity against both the D614G (Figure 3b) and delta (Figure 3c) VoC was determined in a pseudovirus assay. Three months post-vaccination, all but two participants (3%) showed neutralization against D614G with a median qAC50 value of 255 ± 278. Already at this moment, 15 subjects (22%) showed no neutralization of the delta VoC while the overall median qAC50 value was106 ± 72 within the remaining subjects (78%). Furthermore, qAC50 values waned significantly between t_3m_ and t_10m_ with median values of 90 ± 78 for D614G and 50 ± 19 for delta, respectively. It is noteworthy that, at t_10m_, 18 (29%) and 45 (73%) healthcare workers had undetectable neutralization capacity against the D614G and delta VoCs, respectively. Although qAC50 values against D614G and delta VoCs correlated significantly with both anti-S IgG and anti-RBD IgG titers at t_3m_, qAC50 values against both VoCs were no longer significantly correlated with anti-RBD IgG values at t_10m_ (Table 1—row 4–9 and Appendix A). The degree of in vitro RBD–ACE2 binding inhibition (% IH) correlated significantly with qAC50 values against both D614G and delta VoCs at mid- and long-term follow-up (Table 1—row 10–11).

### 3.3. A Minority of Unaffected Individuals Display Circulating RBD-Specific B Cells with Different Kinetics and Correlation to Serology over Time

Furthermore, the temporal evolution of circulating RBD-specific B cells within SARS-CoV-2-naive vaccinated subjects was studied. In a random subset of individuals (*n* = 23) the presence of circulating RBD-specific B cell clones post-vaccination was determined. None of the participants displayed circulating RBD-specific B cells before vaccination (data not shown).

Briefly, four different patterns could be identified within our subset (Figure 4). Six participants (26%, Figure 4 circles) had circulating RBD-specific B cells at both timepoints post-vaccination, while five subjects (22%, Figure 4 squares) did not show any circulating RBD-specific B cells at both timepoints. Notably, only three (13%, Figure 4 upward triangles) and nine subjects (39%, Figure 4 downward triangles) had circulating RBD-specific B cells at t_3m_ or t_10m_, respectively. Lastly, the abundance of RBD-specific B cells correlated significantly with both antibody levels (anti-S IgG and anti-RBD IgG) and functionality at 3 months except for qAC50 values against delta. However, these correlations were no longer significant at 10 months post-vaccination (Appendix A).

### 3.4. Residual SARS-CoV-2-Specific T Cell Activity Is More Retained in CD8^+^ Than CD4^+^ T Cells

To assess whether vaccine-elicited cellular immunity behaves differently compared to humoral immunity, SARS-CoV-2-specific T cell immunity, including (i) cytokine release and (ii) T cell phenotype, was also monitored within this trial. Subsequent analyses were performed within the same random subset of subjects as for the circulating RBD-specific B cells (*n* = 23). Thus, IFN-γ release and membrane activation markers were assessed upon in vitro restimulation of heparinized whole blood with SARS-CoV-2-specific peptides. Of note and similar to humoral immunity, all data from the BTI group were excluded from analyses in this paragraph to reliably visualize the vaccine-specific T cell reactions. Pre-analytical logistic sample issues resulted in the exclusion of data from two subjects.

At all timepoints, a pronounced IFN-γ release was observed in the stimulation control condition for each sample (data not shown). Notably, SARS-CoV-2-specific IFN-γ release non-significantly decreased (*p* = 0.2113; data not shown) between 3 and 10 months post-vaccination with a median t_10m_/t_3m_ ratio of 0.48 and a Q1–Q3 interval of 0.25–0.96 (Figure 5).

Additionally, a flowcytometric assessment of CD69 and CD40L membrane expression within different T cell subsets was performed. The applied gating strategy is shown in Appendix A. Between mid- and long-term follow-up, the level of membrane CD69 expression did not significantly alter (*p* = 0.4380; data not shown) in the T_C_ subset with a median t_10m_/t_3m_ ratio of 0.76 ± 0.36, while this was significantly reduced (*p* = 0.0113; data not shown) in the T_H_ subset with a median t_10m_/t_3m_ ratio of 0.60 ± 0.47. At both sampling moments, there was a significant correlation between CD69 membrane expression in both T_C_ and T_H_ subsets on the one hand, and IFN-γ release on the other hand (Table 2).

Next, upregulation of CD40L membrane expression after restimulation with SARS-CoV-2-specific antigens was examined in T_H_ cells. Membrane CD40L expression levels significantly decreased between t_3m_ and t_10m_ (*p* = 0.0113; data not shown) with a median t_10m_/t_3m_ ratio of 0.26 and a Q1–Q3 interval of 0.12–0.43. Notably, specific IFN-γ release and CD40L^+^ T_H_ cells significantly correlated at t_3m_ but this correlation was no longer significant at t_10m_ (Table 2).

### 3.5. Humoral and Cellular Responses upon BNT162b2 Vaccination Are Not Correlated

Finally, the magnitudes of both humoral and cellular responses at long-term follow-up were aligned for PCA, to reveal the most contributable immunity parameters to the overall variance found within this study (Figure 6). Anti-N IgG titers and qAC50 values against the delta VoC were excluded for PCA since these were below the cut-off index and below the assay-specific LOD, respectively.

No clustering was found in the PC scores plot (Figure 6a). Based on the loadings plot, two large clusters could be identified. These clusters included either all humoral or all cellular immune parameters. However, the rate of RBD-specific B cells is the only parameter that correlated positively with PC 2 at the long-term follow-up test (Figure 6b). At last, combining PC 1 and PC 2, approximately 66% of the overall variance seen within this cohort could be explained (Figure 6c).

### 3.6. Real-World Incidence of Symptomatic BNT162b2 Breakthrough Infections

A total of 13 healthcare workers experienced a RT-PCR-proven BTI within the study timeframe. Detailed demographic information can be found in Appendix A. The age of these participants ranged between 26 and 58 years with a median age of 38 ± 15 years. Additionally, 8 of the 13 subjects were women. BTI occurrence ranged between 44 and 280 days after complete vaccination. From 11 subjects, viral whole genome sequencing was performed on nasopharyngeal swabs that were taken at t_BTI_. Exact sequencing data are available on the GISAID database and revealed that 3 of the 11 subjects were infected with the B.1.117 (i.e., alpha) VoC, whereas the other 8 developed a BTI with the B.1.617.2 (i.e., delta) VoC.

Based on the WHO COVID-19 severity score [39], 3 participants were defined as asymptomatic and 10 as mild. A list of reported symptoms and their intensity can be found in Appendix A. In summary, general feeling of sickness (i.e., malaise) and (dry) cough were the most commonly reported symptoms followed by fever and muscle pain, whereas a sore throat and dyspnea were only rarely reported. The median duration of the symptomatic period was 5 days with a Q1–Q3 interval of 1–7 days. A total of 6 participants took supportive medication during their symptomatic period. Three subjects reported having no knowledge of prior (high-risk) contact with SARS-CoV-2-positive people at time of infection.

### 3.7. Participants with BTI Show Unaltered Cellular Responses but Compromised Humoral Immunity to Vaccination at Mid-Term Follow-Up 

Next, to learn more about possible differences between the vaccine-induced immune response of the 13 participants with reported BTI and the non-BTI group, both responses were compared at mid-term follow-up. Because two participants developed a BTI between t_pre_ and t_3m_, they were excluded from this comparison.

Firstly, no significant differences were detected for both anti-S IgG and anti-RBD IgG titers between the BTI and non-BTI group, whereas significant higher anti-S IgA titers (*p* < 0.01) were found in the non-BTI group. Additionally, no difference in RBD-ACE2 binding inhibition capacity was seen at t_3m_. Furthermore, none of the cellular immune parameters were significantly different at t_3m_ between both groups (Table 3).

Secondly, assessing the immune status at moment of infection (t_BTI_), 7 out of 11 subjects displayed waning antibody responses (e.g., anti-S IgG: Appendix A) and 2 out of 4 subject had decreasing IFN-γ release upon specific restimulation (Appendix A) compared to t_3m_.

### 3.8. Vaccine-Induced SARS-CoV-2 Neutralizing Antibodies Are Not Sufficient to Prevent BTI

Antibodies present in serum of the 11 participants with reported BTI between t_3m_ and t_10m_ were able to inhibit binding of ancestral Wuhan RBD to human ACE2 at t_3m_ with a median % IH of 87.15 ± 12.14. Furthermore, at t_BTI_, 6 out of 11 subjects had a diminished but detectable inhibition capacity compared to t_3m_.

Finally, the VSV pseudovirus qAC50 values against both the D614G and delta VoCs were determined on all timepoints from the 11 affected individuals of which the causative SARS-CoV-2 VoC could be determined via RT-PCR (Figure 7a–l). At t_3m_, qAC50 values against D614G were non-significantly altered between the non-BTI and BTI group with mean qAC50 values of 298 ± 189 and 208 ± 64, respectively. However, the mean qAC50 value against delta was significantly lower (*p* < 0.05, Table 3) in the BTI group (63 ± 13) compared to the non-BTI group (111 ± 55). It is noteworthy that the latter mean qAC50 value only included results from non-BTI individuals who had detectable qAC50 values. All subjects that experienced BTI caused by the alpha VoC had detectable qAC50 values against D614G at time of infection (Figure 7a–c), while only two of the eight subjects with a delta BTI (ID024 and ID041) had remaining activity against the causative VoC at t_BTI_ (Figure 7f,g).

## 4. Discussion

Here, we present the full report of a prospective single-center trial evaluating the sustainability of SARS-CoV-2-specific B and T cell immunity in healthcare workers up to 10 months after receiving two doses of the BNT162b2 vaccine (Figure 1). Interim findings are available at [22]. During the study, multiple SARS-CoV-2 VoCs, including alpha (B1.117), beta (B.1.351), gamma (P1) and delta (B.1.617.2) emerged [24,40,41,42]. Indeed, two SARS-CoV-2 infection waves (from 27 February to 17 June 2021 and from 18 October 2021 to the end of the study) were reported by the Belgian healthcare authorities during the study window. Both infection waves resulted in a substantial increase in both the number of hospital admissions and occupied intensive care beds [43].

Serological findings of this cohort were comparable with those described by other groups that studied serology up to 32 weeks after receiving the BNT162b2 vaccine [44,45,46,47,48,49,50,51,52]. Of note, many of these studies monitored vaccine-induced antibody responses in either patients with multiple pathologies (e.g., multiple sclerosis, hemodialysis) [44,45,46,53,54], SARS-CoV-2 convalescent patients [55,56,57] or between patients receiving one, two or three doses of the BNT162b2 vaccine [57,58]. These different set-ups complicate an accurate comparison between datasets.

In short, all participants of the non-BTI group displayed vaccine-induced anti-S antibodies after vaccination, which undeniably decreased over time with multiple subjects no longer reaching the cut-off index at t_10m_ (Figure 2). In addition, we were not able to retrieve skewing towards either IgG or IgA (Table 1) [59,60,61]. Interestingly and more pronounced to anti-S IgG titers, RBD-directed IgG antibodies were lost in almost half of the participants at t_10m_ (Figure 2), suggesting loss of functional protection by the vaccine-elicited antibodies. Indeed, at long-term follow-up, RBD-ACE2 binding inhibition was clearly reduced with a significant number of subjects (14 of 62) completely losing inhibition capacity compared to mid-term follow-up(Figure 3a). Notably, whereas both anti-S and anti-RBD IgG titers correlated significantly with neutralization at mid-term follow-up_,_ as was also stated by others [59,61,62], only anti-S IgG antibodies remained significantly correlated with neutralization capacity at long-term follow-up (Table 1) [63]. In addition, neutralization against the D614G SARS-CoV-2 VoC was assessed in a more physiological manner using a VSV pseudovirus neutralization assay (Figure 3b). Additionally, anti-S IgG titers remained significantly correlated with D614G-specific activity at both mid- and long-term follow-up, while the correlation with anti-RBD IgG levels was lost at t_10m_ (Table 1). These data surprisingly suggest that anti-S IgG antibodies are possibly better correlates for neutralization. This idea finds further argumentation by reports describing strongly inhibiting non-RBD targeting nAbs [64,65]. However, it should be noted that the loss of correlation with anti-RBD IgG titers could be due to the assay-specific LOD. Moreover, at both mid- and long-term follow-up, we found that functional RBD-ACE2 binding inhibition correlated well with pseudovirus neutralization against D614G (Table 1). These findings are of diagnostic value as automated binding inhibition assay could be used in routine laboratories as a substitute for the more complex VSV pseudovirus neutralization assays. Furthermore, by implementing an approach designed in-house to retrieve RBD-specific B cell clones in SARS-CoV-2 convalescent patients [36], we were able to look beyond antibody secretion and enumerate circulating RBD-specific B cells. The latter can considered to be a surrogate marker for the resident clones present within the secondary lymphoid organs and thus a reflection of ongoing B cell activity [17,66,67,68]. RBD-specific B cells were retrieved in only one third of the study participants at mid-term follow-up [22], contradicting data from another group who have documented antigen-specific clones in virtually all vaccinated subjects [17]. Intriguingly, as more subjects displayed specific clones on long-term follow-up without SARS-CoV-2 infection (Figure 4), it is questionable whether affinity maturation of vaccine-induced antibodies could last that long. The abundance of RBD-specific B cells showed a different PC 2 contribution compared to all other immune parameters at long-term follow-up, implying a different evolution over time [22]. Although serology revealed no sign of infection, it would be highly interesting to dissect whether the long-term observation is vaccine induced rather than due to natural (subinfectious) exposure to viral particles.

Several groups reported SARS-CoV-2-specific IFN-γ release [69,70] in either immunocompromised patients [44,71,72], dialysis patients [48,73] or healthcare workers up to 8 months post-vaccination [10,55,74]. In line with these results, we could confirm SARS-CoV-2-specific IFN-γ release upon in vitro restimulation in all subjects up to 10 months post-vaccination. Interestingly and in contrast to the humoral response, the waning tendency of IFN-y release did not reach significance in the timeframe of this trial, revealing that kinetics of T and B cell activity behave differently. This finds further argumentation since Le Bert et al. showed long-lasting memory of SARS-CoV-2-specific T cell immunity [75]. Moreover, we included T cell phenotyping as innovative secondary read-out to a commercially available IGRA assay, hereby eliminating the need for cumbersome and expensive PBMC isolation, cryopreservation and additional ex vivo restimulation. Between mid- and long-term follow-up, CD69 membrane expression significantly decreased in CD4^+^ T_H_ cells but not significantly in CD8^+^ T_C_ cells (Figure 5), suggesting that the latter subset remained the main producer of IFN-γ. Additionally, membrane-bound CD40L in T_H_ cells significantly dropped between t_3m_ and t_10m_ (Figure 5), rendering these cells less effective in supporting humoral immunity. This aligns with the sharp reduction in specific antibody levels as outlined above. Based on the correlation with expression of T cell activity makers (Table 2), the SARS-CoV-2 IGRA allows us to capture specific T cell activity in a straightforward manner without the need of advanced laboratory equipment.

Thirteen subjects developed a BTI with a SARS-CoV-2 VoC (incidence of 17.11%). Interestingly, Stouten et al. noted a BTI incidence of 4.6% (*n* = 373,070/8,0620,600) in fully vaccinated individuals in Belgium. The researchers calculated a 11 per 100 person years risk to develop a BTI after BNT162b2-based vaccination, which was lower than subjects vaccinated with adeno-vector-based vaccines but higher than those who received the mRNA-1273 vaccine [76]. Moreover, a report evaluating BTI occurrence in the capital region of Denmark (*n* = 1,088,879) mentioned an overall low incidence (hazard ratio of 0.2%) with a tendency towards higher risk to develop BTI with longer time post-vaccination [77]. One might comment that the observed discrepancy in incidence is related to the investigated study population because healthcare workers are more likely to be frequently exposed to infectious doses of viral particles compared to the general population. However, multiple cohorts that monitored BTI incidence in at least 1000 fully vaccinated (i.e., two doses) healthcare workers published incidence rates varying between 0.3 and 1.38% [78,79,80]. On the contrary, one report that compared the influence of a booster vaccine on BTI incidence noted a 21.4% incidence (85/398) within the two-dose regimen control group, which is more similar with the incidence seen in our trial [81]. Additionally, it should be stressed that the majority of BTI within this cohort occurred after 9 months post-vaccination. This can be explained by both the circulation of the more infectious delta VoC coinciding with that timeframe on the one hand and the clearly waning vaccine-induced immunity over time on the other hand. This was confirmed by our observation that less than one-third of the subjects were able to display nAbs against the delta VoC at t_10m_ (Figure 3c). Similar observations were seen in cohorts studied by Katz et al. [82] and Naito et al. [83] who both observed an increase in BTI cases 3 and 5 months post-vaccination, respectively, during increased circulation of SARS-CoV-2 VoCs. In our opinion, this paves the way for an additional booster vaccine to restimulate waning immunity against SARS-CoV-2, especially in immunocompromised patients [84] or the frail elderly population. All 13 subjects reported mild symptoms at time of diagnosis, implying that vaccination protects against severe disease [85,86]. Notably, two subjects with reported BTI showed minimal SARS-CoV-2-specific vaccine-induced humoral and cellular responses at mid-term follow-up (Figure 7d,i) although these healthcare workers did not report compromised overall immunity. Interestingly, at t_3m_, anti-S IgA antibodies were lower in the BTI group, which could be suggestive for less performant mucosal immunity, although this is not the primary objective of intramuscular injection. In line with this, compromised pseudovirus neutralization activity towards the delta VoC was also observed, eventually allowing for symptomatic infection with this VoC (Figure 7l, Table 3). At t_3m_, T cell immunity was not significantly different between the BTI and non-BTI group (Table 3) but, although it was less reduced over time compared to SARS-CoV-2-specific antibodies, apparently this T cell immunity was not sufficient to prevent infection. Upon testing positive via RT-PCR, subjects were recalled for additional sampling to assess their immune status at time of infection (t_BTI_). It is important to point out that differences in both clinical presentation and immune system reactivity between individuals as well as logistic issues in sampling need to be taken into account when addressing these results. At time of infection, individuals affected with the alpha VoC showed diminished but clearly detectable pseudovirus neutralization against D614G, while only two of the eight subjects with a BTI caused by the delta VoC had minimal residual activity against delta (Figure 7f,g). This was in contrast with the study performed by Benning et al. [87], who showed detectable live virus neutralization against the delta VoC up to 8 months after vaccination in 94% of the participants. Additionally, Evans et al. [88] showed detectable but diminishing neutralization capacity against D614G, alpha, beta, delta and omicron VoCs up to 6 months BNT162b2 vaccination. A similar observation was made by a Dutch group who screened 14 BTI cases, caused by either the alpha or delta VoC, and observed that these subjects exerted a vaccine-induced neutralization efficacy against the SARS-CoV-2 VoC [89].

At last, it must be stated that this study has several limitations. Firstly, as the power analysis for this study was primarily based on assessing humoral and cellular immunity, the sample size to monitor BTI is low. However, as only symptomatic infections were further investigated, additional asymptomatic BTI cases might have been missed. Secondly, routine monitoring of exposure parameters including both frequency and viral load could be of high added value in the light of BTI risk stratification. In addition, it is well established that systematic immunity only partially mimics local immunological events in the mucosa. The above-mentioned issues could be addressed by implementing nasopharyngeal swabbing at fixed timepoints for all study participants, but this was beyond the scope of this study. Although we are evolving to a genuine endemic circulation of the SARS-CoV-2 virus, new vaccine escape mutants might appear, warranting further investigation of the sustainability of vaccine-induced systemic immunity as well as local immunity in the upper respiratory tract.

## 5. Conclusions

In this study, we provide an integrated long-term overview of humoral and cellular immunity induced by the BNT16b2 vaccine in healthcare workers in Belgium. Specific immune responses are strongly but not completely reduced over time with different waning patterns between humoral and T cell immunity. In addition, a high incidence rate of symptomatic BTI with SARS-CoV-2 VoCs is reported over time despite only slightly reduced immune responses compared to non-BTI subjects at three months post-vaccination.

## Figures and Tables

**Figure 1 viruses-14-01257-f001:**
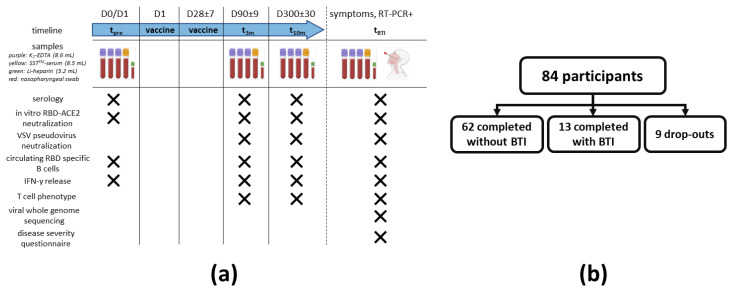
Study design and recruitment. (**a**) Maximum 24 h before receiving prime vaccination, a baseline sampling moment was scheduled (t_pre_). A second and third sampling moment were planned between 80 and 100 days (t_3m_) and 270 and 330 days after baseline (t_10m_), respectively. On each timepoint, serum was collected to assess both SARS-CoV-2 serology and the antibody potency to inhibit RBD-ACE2 interaction (including in vitro and VSV pseudovirus neutralization assays), whereas whole blood from a random subset of participants was used to isolate PBMC to search for circulating RBD-specific B cells and to determine both SARS-CoV-2-specific IFN-γ release and T cell phenotype. When a fully vaccinated participant developed suggestive symptoms or had a prolonged high-risk contact and tested positive via RT-PCR, an additional sampling moment was planned (t_BTI_). At t_BTI_, a disease severity questionnaire was also asked to fill in together with the collection of a nasopharyngeal swab to execute viral whole genome sequencing. (**b**) A total of 84 SARS-CoV-2-naive healthcare workers were enrolled. A schematic overview of the number of dropouts and participants that completed the study either without or with developing a RT-PCR-proven BTI is given. Abbreviations: RBD = viral receptor-binding domain, ACE2 = angiotensin converting enzyme 2 receptor, PBMC = peripheral blood mononuclear cells, IFN-γ = interferon γ, RT-PCR = reverse transcriptase–polymerase chain reaction, VSV = vesicular stomatitis virus, BTI = breakthrough infection. This figure was created using Biorender [31].

**Figure 2 viruses-14-01257-f002:**
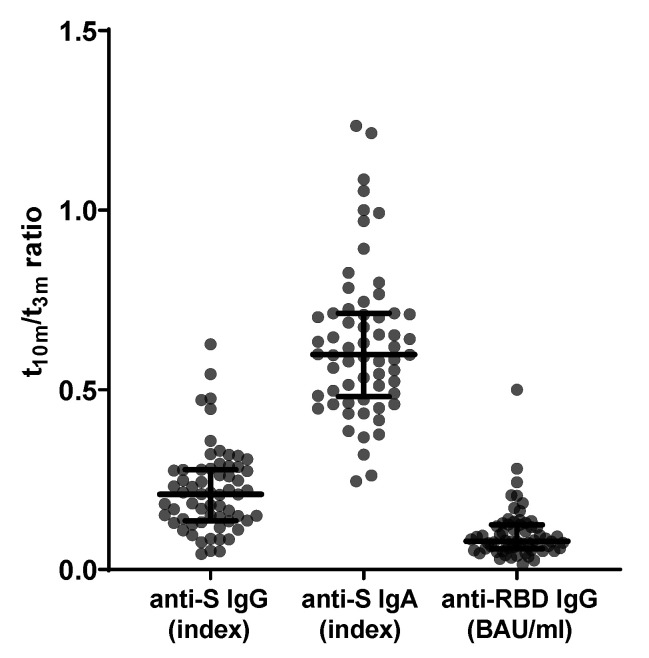
Evolution of SARS-CoV-2-specific serology between mid- and long-term follow-up (*n* = 62). From left to right: anti-S IgG ratio, anti-S IgA ratio and anti-RBD IgG ratio between t_10m_ and t_3m_. Error bars represent median with IQR. Abbreviations: S = spike, RBD = receptor-binding domain, pre = baseline sampling moment before vaccination, 3m = 3 months after baseline, 10m = 10 months after baseline, IQR = interquartile range.

**Figure 3 viruses-14-01257-f003:**
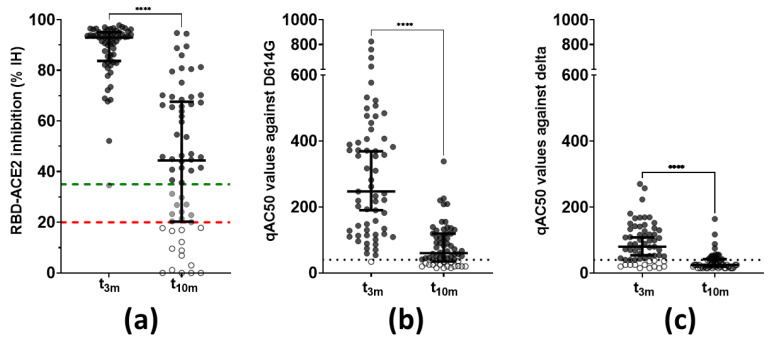
Functionality of vaccine-induced antibodies (*n* = 62). (**a**) Antibody capacity to inhibit the RBD–ACE2 interaction in vitro. This graph includes the full cohort data set with exclusion of patients with BTI, compared to what was shown within the interim analysis of this cohort [22]. Dashed lines = assay-specific cut-offs: above upper (green) line ≥ 35% IH or positive, between upper (green) and lower (red) lines = 20–35% IH or borderline and below lower (red) line ≤ 20% IH or negative. **** Wilcoxon matched-pairs rank test: *p* < 0.0001. (**b**) Pseudovirus neutralization against the D614G VoC. Dotted line = assay-specific LOD. **** Wilcoxon matched-pairs rank test: *p* < 0.0001. (**c**) Pseudovirus neutralization against the delta VoC. Dotted line = assay-specific LOD. **** Wilcoxon matched-pairs rank test: *p* < 0.0001. Error bars represent median with IQR. Abbreviations: % IH = percentage inhibition, RBD = receptor-binding domain, ACE2 = angiotensin converting enzyme 2, qAC50 = ’qualified AC50’: 50% activity against SARS-CoV-2 variant, VoC = variant of concern, 3m = 3 months after baseline, 10m = 10 months after baseline, IQR = interquartile range.

**Figure 4 viruses-14-01257-f004:**
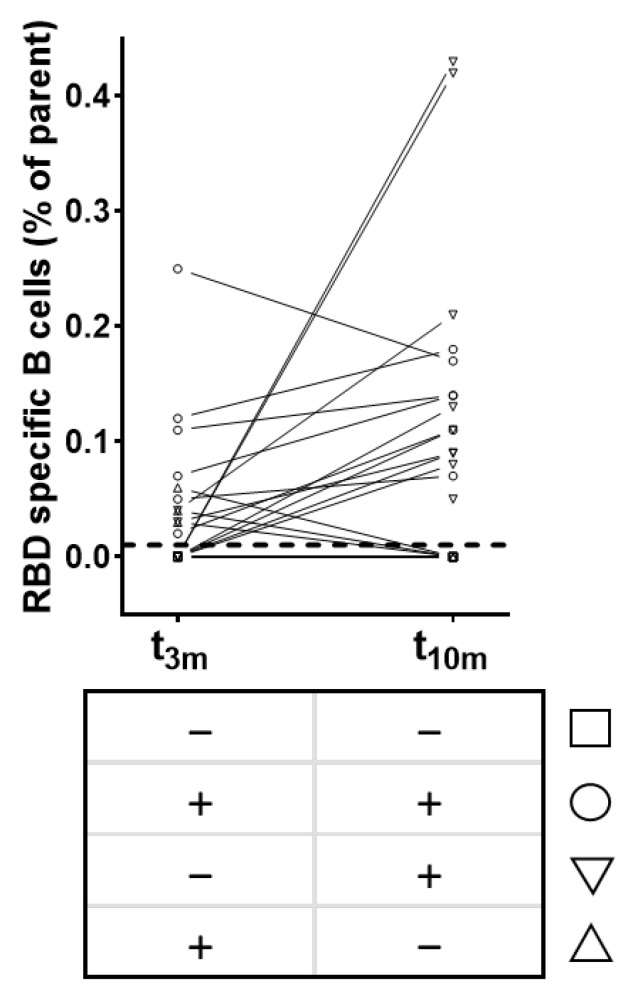
Patterns of circulating RBD-specific B cells after BNT162b2 vaccination (*n* = 23). Squares = subjects without RBD-specific B cells at both t_3m_ and t_10m_. Upward triangles = subjects with RBD-specific B cells at t_3m_ only. Downward triangles = subjects with RBD-specific B cells at t_10m_ only. Circles = subjects with RBD-specific B cells at both t_3m_ and t_10m_. Dashed line = assay-specific cut-off. Abbreviations: RBD = receptor-binding domain, parent = CD3^−^/CD19^+^/Zombie^−^ cells, pre = baseline sampling moment before vaccination, 3m = 3 months after baseline, 10m = 10 months after baseline.

**Figure 5 viruses-14-01257-f005:**
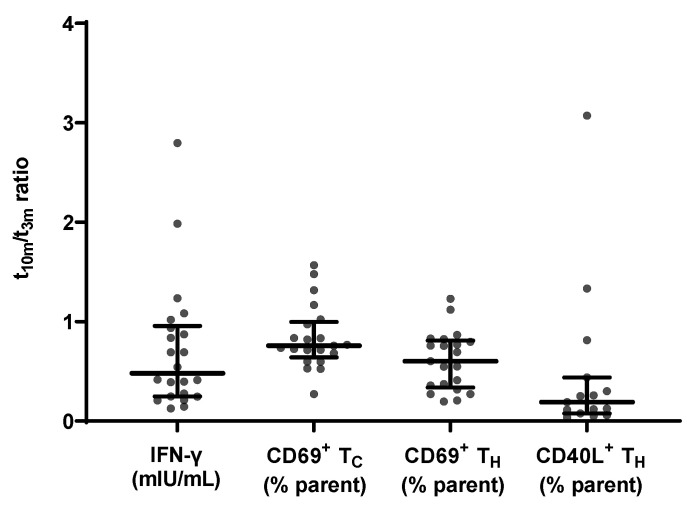
Evolution of SARS-CoV-2-specific cellular immune response parameters measured at mid- and long-term follow-up (*n* = 21). From left to right: specific IFN-γ release ratio, CD69 membrane expression in T_C_ cells ratio, CD69 membrane expression in T_H_ cells ratio and CD40L membrane expression in T_H_ cells ratio between t_10m_ and t_3m_. Error bars represent median with IQR. IFN-γ = interferon γ, pre = baseline sampling moment before vaccination, 3m = 3 months after baseline, 10m = 10 months after baseline, IQR = interquartile range.

**Figure 6 viruses-14-01257-f006:**
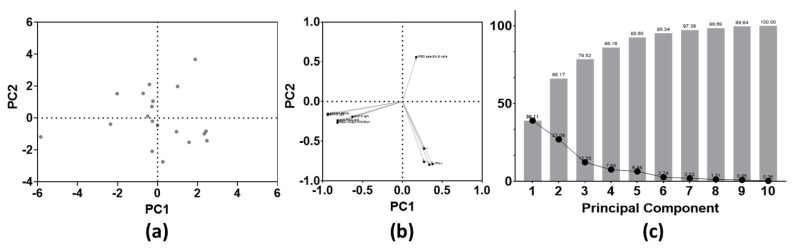
Principal component analysis of both SARS-CoV-2-specific B and T cell parameters at 10 months post-BNT16b2 vaccination (*n* = 10 parameters). (**a**) PC scores plot; (**b**) loadings plot; (**c**) proportion of variance graph. Abbreviations: PCA = principal component analysis, PC = principal component.

**Figure 7 viruses-14-01257-f007:**
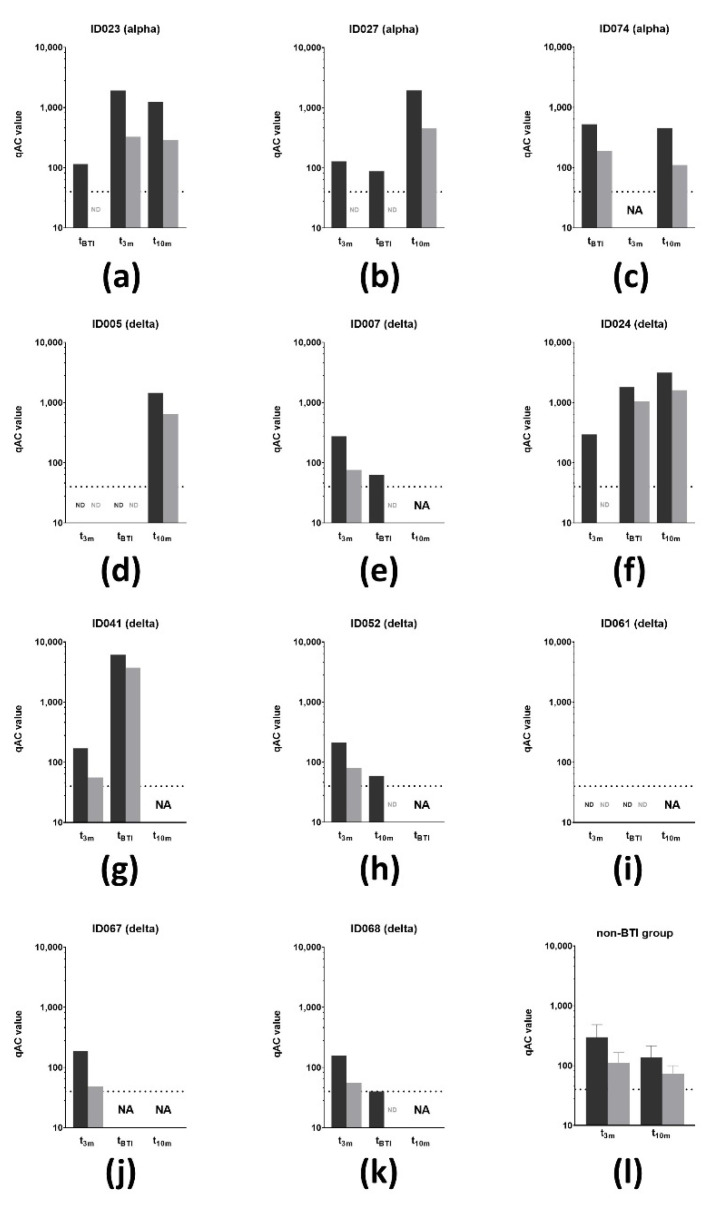
VSV pseudovirus neutralization of SARS-CoV-2 VoC (*n* = 11). (**a**–**c**) qAC50 values (log10) against both D614G (dark grey) and delta (light grey) VoC for three subjects that experienced a BTI caused by the alpha VoC. Two subjects developed a BTI before t_3m_, while the other subject developed this between t_3m_ and t_10m_. (**d**–**k**) qAC50 values (log10) against both D614G (dark grey) and delta (light grey) VoCs for eight subjects that experienced a BTI caused by the delta. (**l**) Mean qAC50 values against both D614G (dark grey) and delta (light grey) VoCs of the non-BTI group at mid- and long-term follow-up. Dotted line = assay-specific LOD. Importantly, data points below the LOD were not included in the graph and thus were excluded for statistical analyses. Error bars represent mean with SD. Abbreviations: VSV = vesicular stomatitis virus, VoCs = variants of concern, BTI = breakthrough infection. qAC50 = ’qualified AC50’: 50% activity against SARS-CoV-2 variant, NA = not available, ND = not detectable (i.e., below assay-specific LOD), 3m = 3 months after baseline, 10m = 10 months after baseline, LOD = limit of detection, SD = standard deviation.

**Table 1 viruses-14-01257-t001:** Correlation of SARS-CoV-2-specific humoral parameters measured after BNT162b2 vaccination.

Parameter VS. Parameter	t_3m_		t_10m_	
	Spearman r (*p*-Value)	Pearson R²	Spearman r (*p*-Value)	Pearson R²
Anti-S IgG vs. anti-S IgA	0.6218 (*p* < 0.0001)	0.3438	0.4803 (*p* < 0.0001)	0.1854
Anti-S IgG vs. anti-RBD IgG	0.8988 (*p* < 0.0001)	0.6325	0.5074 (*p* = 0.005)	0.5295
Anti-S IgA vs. anti-RBD IgG	0.5502 (*p* < 0.0001)	0.2162	0.3728 (*p* = 0.0464)	0.0067
Anti-S IgG vs. RBD-ACE2% IH	0.8421 (*p* < 0.0001)	0.5941	0.9162 (*p* < 0.0001)	0.7336
Anti-S IgG vs. qAC50 D614G	0.8903 (*p* < 0.0001)	0.6358	0.8486 (*p* < 0.0001)	0.6496
Anti-S IgG vs. qAC50 delta	0.7462 (*p* < 0.0001)	0.3699	0.6357 (*p* = 0.0046)	0.1802
Anti-RBD IgG vs. RBD-ACE2% IH	0.8194 (*p* < 0.0001)	0.3997	0.4793 (*p* = 0.0085)	0.1352
Anti-RBD IgG vs. qAC50 D614G	0.8127 (*p* < 0.0001)	0.5336	0.3210 (*p* = 0.1025)	0.1893
Anti-RBD IgG vs. qAC50 delta	0.6784 (*p* < 0.0001)	0.3757	0.4259 (*p* = 0.1007)	0.0178
RBD-ACE2% IH vs. qAC50 D614G	0.8534 (*p* < 0.0001)	0.4578	0.8853 (*p* < 0.0001)	0.6765
RBD-ACE2% IH vs. qAC50 delta	0.8087 (*p* < 0.0001)	0.4200	0.8475 (*p* < 0.0001)	0.3844

Abbreviations: S = spike, RBD = receptor-binding domain, ACE2 = angiotensin converting enzyme 2, % IH = percentage inhibition, qAC50 = ’qualified AC50’: 50% activity against SARS-CoV-2 variant, 3m = 3 months after baseline, 10m = 10 months after baseline.

**Table 2 viruses-14-01257-t002:** Correlation of SARS-CoV-2-specific cellular parameters measured after BNT162b2 vaccination.

Parameter VS. Parameter	t_3m_		t_10m_	
	Spearman r (*p*-Value)	Pearson R²	Spearman r (*p*-Value)	Pearson R²
IFN-γ vs. CD69^+^ T_C_ cells	0.7649 (*p* < 0.0001)	0.3891	0.5030 (*p* = 0.0144)	0.2360
IFN-γ vs. CD69^+^ T_H_ cells	0.6961 (*p* = 0.0005)	0.6568	0.7915 (*p* < 0.0001)	0.5962
IFN-γ vs. CD40L^+^ T_H_ cells	0.6171 (*p* = 0.0029)	0.6344	0.4070 (*p* = 0.0539)	0.0898

Abbreviations: IFN-γ = interferon γ, 3m = 3 months after baseline, 10m = 10 months after baseline.

**Table 3 viruses-14-01257-t003:** Comparison of both SARS-CoV-2-specific humoral and cellular parameters between the BTI and non-BTI group measured 3 months after BNT162b2 vaccination.

Immune Parameter	t_3m_		
	BTI Group (Mean ± SD) *	Non-BTI Group (Mean ± SD) *	*p*–Values **
Humoral immune response			
Anti-S IgG (index)	21.83 ± 11.33	26.79 ± 8.508	*p* = 0.2418
Anti-S IgA (index)	1.740 ± 0.959	2.948 ± 1.492	*p* = 0.0077
Anti-RBD IgG (BAU/mL)	773.2 ± 245.4	981.6 ± 555.0	*p* = 0.5883
RBD-ACE2 inhibition (% IH)	75.99 ± 32.58	88.19 ± 11.37	*p* = 0.1888
qAC50 D614G VoC	208.1 ± 64.28	297.7 ± 186.8	*p* = 0.3213
qAC50 delta VoC	62.86 ± 13.08	110.6 ± 54.69	*p* = 0.0173
Cellular immune response			
IFN-γ (mIU/mL)	1349 ± 936.7	1950 ± 2094	*p* = 0.7394
CD69+ T_C_ cells (% of parent)	17.64 ± 7.325	18.70 ± 8.107	*p* = 0.8501
CD69^+^ T_H_ cells (% of parent)	11.96 ± 6.834	16.87 ± 7.717	*p* = 0.1570
CD40L^+^ T_H_ cells (% of parent)	0.476 ± 0.451	0.551 ± 0.571	*p* = 0.6058

* Sample sizes. BTI group: *n* = 10–12 for the humoral parameters and *n* = 4 for the cellular parameters. Non-BTI group: *n* = 62 for the humoral parameters except when there was an undetectable signal and *n* = 23 for the cellular parameters. ** Mann–Whitney U test. Abbreviations: BTI = breakthrough infection, SD = standard deviation, S = spike, RBD = receptor-binding domain, ACE2 = angiotensin converting enzyme 2, qAC50 = ’qualified AC50’: 50% activity against SARS-CoV-2 variant, IFN-γ = interferon γ.

## Data Availability

Sequencing data: see Appendix A for all GISAID database ID numbers.

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
