# Peer review of "High Incidence of SARS-CoV-2 Variant of Concern Breakthrough Infections Despite Residual Humoral and Cellular Immunity Induced by BNT162b2 Vaccination in Healthcare Workers: A Long-Term Follow-Up Study in Belgium"

_viruses, 2022, doi:10.3390/v14061257_

Round 1

Reviewer 1 Report

Very interesting paper.

Author Response

Dear independent reviewer #1. 

We thank you for reviewing our work and were pleased to read that you found it interesting. 

Reviewer 2 Report

• What are the inclusion and exclusion criteria in the study?  • Which randomization method was used in the distribution of the individuals included in the study to the groups?  • Which blinding (masking) method was used in the study?  • Data analysis or Statistical analysis sub-section title should be added to the Materials and Methods.  • How was the sample size determined? This information should be explained in the Materials and Methods section.  • Which sampling (probable or non-probable, etc.) method was used in the study?  • Statistical tests for hypothesis testing and their assumptions should be specified in the statistical analysis of the study in the Materials and Methods section.  • The details (version, license number, etc.) of the statistical package(s) or program(s) should be given in the section of "Data Analysis or Statistical Analysis".  • It should be explained how the qualitative and quantitative data are summarized under the sub-heading of Statistical Analyzes in the Materials and Methods section of the study.  • The exact P values should be added to the table(s) (p=0.25; p=0.03).  • Which methods are used to model relationships between variables?  • The descriptions and other descriptive values/data should be defined on the tables and shapes. • Are the data subjected to pre-processing?  • How were extreme/outlier values in the data determined and resolved?

Author Response

Dear independent reviewer #2,

We are pleased to receive valuable and critical feedback from you and want to thank you for your questions/suggestions. Please find  all comments addressed point-by-point in attachment (word-doc).

Reviewer 3 Report

Summary of the strength and Weakness of the study

This study by Calcoen et. al longitudinally evaluated the BNT162b2 pfizer mRNA vaccine induced immune responses in the Belgium healthcare workers up to 10 months. Authors reported a high incidence of SARS-CoV-2 break through infections despite presence of both residual humoral and cellular immunity induced by BNT162b2 vaccination.

The experimental methods, results and findings of this study are interesting and convincing. This study is contributing additional information to the area of understanding vaccine induced immunity in SARS-CoV-2 infection. Limitations of the study are also well described. However, the manuscript is lengthy and too much detailed writing is diluting the major outcomes and focus of this study. Authors should focus on conveying their findings in more succinct way of writing. Further, this study can be improved and bring more value if, a clarity of step by step process of the methods can be incorporated specially related to RBD-ACE2 inhibition assays, T cells analysis and IFN-g assays, missing kit cat. Numbers throughout the methods.

Though authors have evaluated RBD and S specific Ab responses over time. It is known that because of RBD mutations especially in the RBM regions majority of the Ab response decline and hence increase the incidence break through infections due to lack of protective immunity. It would be interesting to determine S2-domain specific immune responses over time up to 10 months, which is highly conserved among all human CoVs and target of broadly neutralizing antibodies.

Minor comments:

Line 61 and 69: Please correct the name of Moderna mRNA vaccine, it should be mRNA-1273, in place of mrNA-1732/172

Author Response

Dear independent reviewer #3,

We want to thank you for the constructive and valuable feedback on our paper. Please find our answers on your questions/suggestions in attachment (word-doc). 

Round 2

Reviewer 3 Report

Thank you for your responses, everything looks good overall.